# Myo5b Transports Fibronectin-Containing Vesicles and Facilitates FN1 Secretion from Human Pleural Mesothelial Cells

**DOI:** 10.3390/ijms23094823

**Published:** 2022-04-27

**Authors:** Tsuyoshi Sakai, Young-yeon Choo, Osamu Sato, Reiko Ikebe, Ann Jeffers, Steven Idell, Torry Tucker, Mitsuo Ikebe

**Affiliations:** Department of Cellular and Molecular Biology, The University of Texas Health Science Center at Tyler, 11937 US Highway 271, Tyler, TX 75708-3154, USA; tsuyoshi.sakai@uthct.edu (T.S.); yungyeon.choo@uthct.edu (Y.-y.C.); osamu.sato@uthct.edu (O.S.); reiko.ikebe@uthct.edu (R.I.); ann.jeffers@uthct.edu (A.J.); steven.idell@uthct.edu (S.I.); torry.tucker@uthct.edu (T.T.)

**Keywords:** fibronectin, myosin 5b, fibrosis, mesothelial cell, vesicle transport, secretion, motor protein

## Abstract

Pleural mesothelial cells (PMCs) play a central role in the progression of pleural fibrosis. As pleural injury progresses to fibrosis, PMCs transition to mesenchymal myofibroblast via mesothelial mesenchymal transition (MesoMT), and produce extracellular matrix (ECM) proteins including collagen and fibronectin (FN1). FN1 plays an important role in ECM maturation and facilitates ECM-myofibroblast interaction, thus facilitating fibrosis. However, the mechanism of FN1 secretion is poorly understood. We report here that myosin 5b (Myo5b) plays a critical role in the transportation and secretion of FN1 from human pleural mesothelial cells (HPMCs). TGF-β significantly increased the expression and secretion of FN1 from HPMCs and facilitates the close association of Myo5B with FN1 and Rab11b. Moreover, Myo5b directly binds to GTP bound Rab11b (Rab11b-GTP) but not GDP bound Rab11b. Myo5b or Rab11b knockdown via siRNA significantly attenuated the secretion of FN1 without changing FN1 expression. TGF-β also induced Rab11b-GTP formation, and Rab11b-GTP but not Rab11b-GDP significantly activated the actin-activated ATPase activity of Myo5B. Live cell imaging revealed that Myo5b- and FN1-containing vesicles continuously moved together in a single direction. These results support that Myo5b and Rab11b play an important role in FN1 transportation and secretion from HPMCs, and consequently may contribute to the development of pleural fibrosis.

## 1. Introduction

Pleural mesothelial cells (PMCs) form a cellular monolayer that covers the chest wall and the lung forming the parietal and visceral pleural, respectively. They have multiple intercellular adherens junctions and focal adhesions that anchor the cells to the extracellular membrane. However, PMCs change their phenotype from an epithelioid phenotype to more profibrotic phenotype in response to cytokines released during pleural inflammation during the development of fibrosis [1]. During transition, mesothelial cells begin expressing α-smooth muscle actin (α-SMA) and demonstrate cytoskeletal reorganization and phenotypic changes to become myofibroblast. PMC derived myofibroblasts also demonstrate the expression and secretion of extracellular matrix (ECM) proteins such as collagen 1 (Col-1), which is directly involved in pleural fibrosis. As described above, another key ECM protein that is supposed to be involved in the development of pleural fibrosis is fibronectin, yet the role of PMCs in the production and deposition of fibronectin in the progression of pleural fibrosis is poorly understood.

Fibrotic tissue is characterized by the excessive accumulation of cells and extracellular matrix proteins. Col-1 plays a central role in the development of fibrosis. Col-1 fibrils show disorganized structure due to cross-linking in fibrosis with increased collagen cross-linking enzymes such as lysyl oxidase (LOX) [2,3]. Another important ECM protein during the development of fibrosis is fibronectin (FN1). FN1 mediates the interaction between various ECM components and cells [4] and assembly of other ECM components [5]. For instance, FN1 matrix is essential for formation of collagen network [6]. FN1, specifically the splice variant containing the type III extradomain (EDA), is increased during fibrosis and plays an important role in myofibroblast differentiation by TGF-β [7]. Further, EDA neutralizing antibodies and peptide inhibitor inhibits TGF-β 1 activation [8,9]. Conversely, increased expression of the EDA isoform is associated with tissue remodeling [10]. ECM and myofibroblasts influence each other and it has been suggested that FN1-EDA plays a positive role in activating TGF-β signaling [8,11,12], and that TGF-β regulates FN1-EDA expression during myofibroblast differentiation [13]. 

The mechanism of fibronectin transportation and secretion and the identity of motor proteins responsible for these processes are unknown. Actin-based motor systems are reported to play a key role in the directional transportation of intracellular vesicle movement [14,15,16]. Myosin is an actin-based motor protein, which constitutes a super family of more than 35 members and plays various roles in diverse cell motility, contractility and directional cargo transportation [17]. Myosin family members can be classified into two groups based upon their motor properties. One is non-processive or conventional myosin, such as class II myosin, that binds to actin a small portion of the time during a cross-bridge cycle or ATP hydrolysis cycle, thus spending a majority of time dissociated from actin. This type of myosin is suitable for the production of force. The other type of myosin associates with actin for a majority of time during a cross-bridge cycle, thus allowing it to move it processively on actin filaments. The latter type of myosins is thought to be suitable for intracellular vesicular movements [18]. Class V myosins (myosin V) are best studied processive myosin and capable to transport intracellular vesicles [19]. In the case of melanocytes, Myo5a, an isoform of class V myosin in myosin superfamily, transports melanosomes [20]. Based on their functions, we predicted that one of class V myosin would be capable of transporting and secreting vesicular fibronectin.

Our objective is to elucidate the production and the secretion mechanism of fibronectin from human pleural mesothelial cells (HPMCs). We hypothesized that PMCs play a critical role in fibronectin production and that fibronectin-containing vesicles are transported from the perinuclear domain of the cellular periphery through specific motor proteins rather than by simple diffusion. This process would facilitate efficient secretion of fibronectin from PMCs, thus promoting maturation of ECM structure, fibroblast differentiation and subsequently fibrosis. In the present study, we attempted to identify the transportation mechanism responsible for fibronectin secretion and deposition into the pleural space using advanced molecular and cellular imaging approaches. We found that Myo5b transports fibronectin and facilitates secretion from HPMCs. It is expected that this work will contribute to understanding of the cellular and molecular mechanisms of fibronectin secretion, thus providing clues to understand pleural injuries and the development of novel therapies for pleural fibrosis.

## 2. Results

### 2.1. Fibronectin (FN1) Expression Is Increased in the Visceral Pleura of Carbon Black Bleomycin(CBB)-Induced Fibrosis and S. pneumoniae-Induced Empyema Mouse Models

We and others have shown that profibrotic stimuli such as TGF-β induce distinct phenotypic changes of human pleural mesothelial cells (HPMCs). Because of the similarity to epithelial-mesenchymal transition (EMT) this process has been termed mesothelial mesenchymal transition (MesoMT) [21,22,23,24,25]. During MesoMT, there is a significant increase in α-smooth muscle actin (αSMA) expression and the expression of extracellular matrix (ECM) proteins, such as collagen-1 (Col-1), by HPMCs [26,27]. Similar findings were observed in vivo. However, FN1 expression in HPMCs and in the pleura during the development of pleural fibrosis has not been characterized. To address this gap in knowledge, we first examined the deposition of FN1 in the pleura of mice with carbon black bleomycin (CBB) induced pleural fibrosis. Immunofluorescent analysis of αSMA, a marker of MesoMT, showed a marked increase in αSMA expression in the thickened pleural rind of CBB injured mice compared to saline-treated control mice (Figure 1A). FN1 and (Col-1) expression were likewise enhanced in the visceral pleura of CBB injured mice (Figure 1B). FN1 deposition was also observed in the thickened pleura of mice having *S. pneumoniae*-induced empyema (Figure 1C). Further, FN1 deposition was colocalized with the (Col-1) in the thickened pleural rind of *S. pneumoniae* injured mice.

### 2.2. TGF-β Increases Expression and Secretion of FN1 by HPMCs

We and others have shown that MesoMT (mesothelial mesenchymal transition) of pleural mesothelial cells contributes to the enhanced deposition of collagen in injured pleural tissues [21,28,29,30]. Therefore, we next asked whether MesoMT contributes to FN1 deposition in pleural tissues. To address this question, we treated primary human pleural mesothelial cells (HPMCs) with TGF-β to induce MesoMT. RNA was then isolated from these cells and analyzed by qPCR. As shown in Figure 2A, TGF-β significantly (*p* < 0.001) up-regulated FN1 mRNA expression, as well as αSMA, in HPMCs. TGF-β treated lysates and conditioned media were next examined for the expression and secretion of FN1. The qPCR results were confirmed by Western blot analysis that showed significantly increased FN1 protein expression with TGF-β stimulation (Figure 2B,C). We also found that FN1 was significantly increased in the cell culture supernatant of TGF-β treated HPMCs compared to controls (Figure 2B,C). It should be noted that the increase in the expression and secretion of FN1(EDA) isoform that has been shown to be related to myofibroblast activation during fibrosis [7] is more prominent (Figure 2B,C). These results suggest that TGF-β stimulation not only increases mRNA and protein expression of FN1, but also enhances FN1 secretion from HPMCs. Further, these data suggest that HPMCs directly contribute to the deposition of FN1 in injured pleural tissues.

### 2.3. Myo5b Co-Localizes with FN1 in HPMCs

Because TGF-β increased the expression and secretion of FN1 from HPMCs, we hypothesized that the directional transportation system of FN1-containing vesicles in HPMC is likewise activated by TGF-β stimulation. Figure 3A shows co-localization of FN1 with actin structures in HPMCs. Co-localization analysis using confocal imaging of FN1 with F-actin revealed significant co-localization of FN1 along the line of filamentous structure of F-actin (Figure 3A). The result suggests that FN1-containing vesicles are transported along with actin filaments presumably by means of actin-based motor protein, myosin. Myosin constitutes a large superfamily, and certain classes of myosin family members are suitable motors for cargo transportation because they can associate with the cargo molecules for a prolonged time, and these myosin motors are often called “processive myosin”. Class V myosin is the best known processive myosin that can transport the associated cargo molecules for long distances. Among the class V myosin, myosin Va and myosin Vb are processive myosin [19,31,32]. As shown in Figure 3A, a notable portion of endogenous Myo5b co-localized with FN1-containing vesicles. Further, the colocalized signals were found on actin filaments. Conversely, Myo5a did not show appreciable colocalization with FN1 (Figure 3C).

To further study the close association of Myo5b and FN1 induced by TGF-β stimulation, we next performed a proximity ligation assay. This assay can detect close association between the two proteins within 50 nm distance [33,34]. As shown in Figure 4A,B, the proximity ligation signal between Myo5b and FN1 was significantly (*p* < 0.001) increased by TGF-β stimulation compared to PBS controls. Moreover, the signals were found on actin filaments, indicated by arrow heads in Figure 4A (right). These results suggest that TGF-β stimulation induces the association of Myo5b with the FN1-containing vesicles.

### 2.4. Myo5b Gene Silencing Attenuates Secretion of FN1 from HPMCs

Since Myo5b associates with FN1-containing vesicles upon TGF-β stimulation, it is plausible that Myo5b is involved in FN1 transportation and secretion. We next determined the role of Myo5b in the transportation/secretion of FN1 in HPMCs. We down-regulated Myo5b expression with targeting siRNA and examined the effect of gene silencing of Myo5b on FN1 secretion. Quantitative PCR analysis revealed that Myo5b siRNA significantly reduced the Myo5b mRNA level both in the presence and absence of TGF-β (KD efficiency; 95.5% before TGF-β, 96.5% after TGF-β), suggesting the Myo5b siRNA effectively diminished transcription of myosin 5B (Figure 5A). Myo5b protein expression was likewise significantly diminished by the siRNA (Figure 5B). We next examined the effect of Myo5b downregulation on FN1 secretion. FN1 in culture supernatant was significantly reduced by the Myo5b knockdown (Figure 5C,D). Since siRNA for Myo5a and Myo5b did not affect the mRNA level of FN1 (Figure 5A), the result suggests that gene silencing of Myo5b diminished the secretion process of FN1.

On the other hand, Myo5a KD did not show the significant changes in FN in culture supernatant (Figure 5C,D), while it effectively diminished the mRNA (Figure 5A) and protein expression of Myo5b (Figure 5B). These results suggest that Myo5b, but not Myo5a, is important for the secretion of FN1-containing vesicles, presumably transportation of the vesicles containing FN1 to facilitate secretion.

### 2.5. Rab11b Gene Silencing Attenuates Secretion of FN1 from HPMCs

Rab family proteins have been known to link vesicular trafficking proteins to specific vesicles [35,36,37,38,39]. Further, it was reported that Myo5b associates with Rab11a-containing vesicles in Hela and bladder umbrella cells [40,41,42]. As such, we next examined whether Rab11 plays a role in FN1 secretion.

To address this notion, we first examined the effect of gene silencing of Rab11 on FN1 secretion. Rab11b siRNA significantly reduced Rab11b mRNA in the presence and absence of TGF-β (Figure 6A, KD efficiency; 94.3 % before, 93.6% after TGF-β), whereas FN1 mRNA were unchanged by the Rab11b siRNA (Figure 6A). Likewise, Rab11a siRNA significantly reduced Rab11a mRNA before and after TGF-β stimulation (KD efficiency; 94.4 % before, 92.7% after TGF-β) without changing FN1 mRNA level (Figure 6A). Western blot analysis of total cell lysates revealed that Rab11b specific siRNA effectively diminished the protein expression of Rab11b. Likewise, Rab11a specific siRNA effectively diminished the protein expression of Rab11a (Figure 6B). It should be noted that FN1 expression was not affected by Rab11b siRNA or Rab11a siRNA.

Using these siRNAs, we next examined the effect of Rab11b gene silencing on FN1 secretion. While FN1 protein in cell lysates were not significantly altered by the Rab11b gene silencing, FN1 in culture supernatant was significantly reduced by the Rab11b siRNA (Figure 6C,D). Conversely, Rab11a knock-down did not result in the significant decrease in the FN1 in the culture supernatants (Figure 6C,D). These results suggest that Rab11b, but not Rab11a, plays a role in the secretion of FN1-containing vesicles, presumably by bridging Myo5b and FN1-containing vesicles.

### 2.6. TGF-β Induces Association of Myo5b with Rab11b and FN1-Containing Vesicles in HPMCs

We next determined whether TGF-β facilitates the association of Myo5b to FN1-containing vesicles via Rab11b for transportation of the vesicles. We first examined whether FN1-containing vesicles colocalize with Rab11b and Myo5b. HPMCs were co-infected with vectors expressing GFP-Myo5b, mCherry-FN1, and V5-Rab11b (Figure 7). Infected cells were examined for the localization of these proteins in HPMCs after TGF-β stimulation. V5-Rab11b showed notable colocalization with GFP- Myo5b and mCherry-FN1 (Figure 7).

To further study the effect of TGF-β on the close association of Myo5b and Rab11b, we again performed a proximity ligation assay that can detect close association between the two proteins within 50 nm distance. As shown in Figure 8A, the proximity ligation signal between Myo5b and Rab11b was significantly increased by TGF-β stimulation compared to PBS controls. These results suggest that TGF-β stimulation induces the association of Myo5b with Rab11b-containing vesicles. The proximity ligation signal between FN1 and Rab11b was also significantly increased after TGF-β stimulation (Figure 8B). Our results suggest that Myo5b resides on FN1-containing vesicles through the binding to Rab11b after TGF-β treatment. This implies that Rab11b is required for Myo5b to transport FN1-containing vesicles.

### 2.7. Activation of Rab11b Induces Association with Myo5b and Facilitates FN1 Secretion

We next asked if the activation of Rab11b facilitates the association of Myo5b with FN1-containing vesicles. Further, if TGF-β induces the activation of Rab11b, thus promoting the association of Myo5b with FN1-containing vesicles for transportation and secretion. Figure 9 shows the colocalization between GFP-Myo5b and V5-Rab11b wild type, V5-Rab11b S25V (dominant negative mutant), and V5-Rab11b S20V (active mutant) in the presence of TGF-β, respectively. The active mutant of Rab11b (S20V) showed punctate localization that was highly colocalized with GFP-Myo5b (Figure 9). On the other hand, staining of the inactive mutant of Rab11b (S25V) was rather diffuse. Myo5b also showed diffuse localization in the cells expressing Rab11b (S25V). These results suggest that Myo5b associate with vesicles via interaction with the active form of Rab11b. V5-Rab11b wild type also showed notable colocalization with Myo5b (Figure 9 Top panel). This suggests that WT Rab11b is partially activated in the cells after TGF-β stimulation.

To further confirm the effect of Rab11b activation on the binding to Myo5b, we performed a Proximity Ligation Assay. V5-Rab11b variants were expressed in HPMCs along with GFP- Myo5b and subjected to Proximity Ligation assay. HPMCs expressing similar levels of the three V5-Rab11b variants were subjected to the analysis (Figure 10A). Figure 10B shows the proximity ligation signal between V5-Rab11b variants and GFP- Myo5b in the presence of TGF-β. The signal intensity was significantly lower for the dominant negative mutant (S25V) compared to the active mutant (S20V). On the other hand, signal intensity of the wild type (WT) was significantly higher than the negative mutant (S25N) but lower than the active mutant (V5-Rab11b S20V) (Figure 10B,C). The result suggests that Rab11b wild type is partially activated in TGF-β treated cells, thus associating with Myo5b. The results also suggest that Rab11b associates with FN1-containing vesicles when it forms an active conformation. To test this hypothesis, the role of Rab11b activation on Myo5b binding was examined (Figure 11).

We purified Rab11b and the tail domain of Myo5b, a cargo binding domain (see Materials and Methods) and performed the direct binding assay. The isolated Rab11b was incubated with either GTPγS or GDPβS to produce the active and inactive form, respectively. The mixtures were then subjected to GST-pull-down assay with GST- Myo5b tail. As shown in Figure 11A, GTPγS markedly facilitated the binding of Rab11b to Myo5b tail, indicating that the activation of Rab11b is critical for the binding of Rab11b to Myo5b. The result was consistent with the imaging analysis shown in Figure 9 and Figure 10, and supports our hypothesis that Myo5b binds to Rab11b, when activated. Next, we determined whether TGF-β activates Rab11b in HPMCs. To monitor the activation of Rab11b, we used the Rab11 binding domain of Rab11FIP3 (FIP3-RBD) as a probe [43,44]. Figure 11B shows the effect of GTPγS and GDPβS on the binding of Rab11b to FIP-RBD. GTPγS markedly enhanced the binding of Rab11b to GST-FIP3-RBD. Using FIP3-RBD as a probe, we next examined the effect of TGF-β stimulation on the activation of Rab11b in HPMCs. Flag-Rab11b was added to the lysates of HPMCs treated with or without TGF-β, and subjected to GST-pull-down assay (Figure 12C). TGF-β stimulated HPMCs showed a notable increase in the binding with GST-FIP3-RBD compared to PBS treated controls. The result suggests that TGF-β activates Rab11b in HPMCs to form Rab11b-GTP, thus facilitating the binding to Myo5b.

### 2.8. Rab11 Activates the Motor Function of Myo5b

Our results suggest that Myo5b binds to the active form of Rab11b, i.e., Rab11b-GTP, and facilitates the binding to FN1-containing vesicles that reside with Rab11b.

We next asked whether the active Rab11b binding to Myo5b promotes the motor function of Myo5b. To address this notion, we examined the effect of Rab11b on the motor activity of Myo5b. Full-length Myo5b was isolated and the effect of Rab11b on the motor function was determined by measuring the actin activated ATPase activity of Myo5b. Rab11b with GTPγS highly activated the actin-activated ATPase activity, while Rab11b-GDPβS maginally activated the ATPase activity (Figure 12). Since the actin-activated ATPase activity is coupled with the cross-bridge cycling activity, i.e., motor cycling [45], the result indicated that the active form of Rab11b activates the motor function of Myo5b in addition to facilitating the association of Myo5b to the FN1-containing vesicles.

### 2.9. Direct Visualization of Movement of FN by Myo5b in Living Cells

To obtain conclusive evidence that Myo5b transports FN1-containing vesicles, we performed live cell imaging of the movement of FN1-containing vesicles with Myo5b. HPMCs were co-transduced with mCherry-FN1(EDA) expressing viral vector and GFP- Myo5b expressing viral vector. GFP-Myo5b showed punctate localization in HPMCs with TGF-β stimulation (Figure 13). To analyze Myo5b dependent movement of FN1-containing vesicles, we monitored time projection images. The images clearly demonstrated the directional movement of FN1-containing vesicles in live HPMCs stimulated by TGF-β (Figure 13A). The trajectory images of mCherry-FN1 and GFP-Myo5b, clearly showed the co-movement of mCherry-FN1 and GFP-Myo5b (Figure 13B,C and Appendix A). We next performed Kymograph analysis of the movement that represents the time course of the directional movement of GFP- Myo5b and mCherry-FN1(EDA). The results show that mCherry-FN1(EDA) and GFP-Myo5b continuously move together without dissociation from the track (Figure 13D). The Myo5b/FN1 complex moved with a velocity of 0.26 μm/sec, stopped, and in the same direction towards cell peripheries (Figure 13).

## 3. Discussion

It has been known that FN1, especially the FN1 EDA variant is highly up-regulated during differentiation of myofibroblasts, and plays an important role in myofibroblast activation and differentiation [11,13]. Moreover, FN1 plays a critical role in ECM assembly [46,47], and collagen network organization. FN1 plays an essential role in the interaction between ECM and myofibroblast, which can influence TGF-β signaling and phenotypic changes of myofibroblasts, yet the mechanism of FN1 secretion from myofibroblasts is poorly understood. Further, as a component of the neomatrix, FN1 abundance may impact resolution of pleural loculations. As such the identification of processes that may regulate the expression and secretion of neomatrix components could affect the effectiveness of therapeutics which promote dissolution of pleural loculations [48]. In the present study, we attempted to clarify the mechanism of the transportation and secretion of FN1 from myofibroblasts that is differentiated from mesothelial cells through mesothelial-mesenchymal transition (MesoMT) [24,49].

The present study is the first to identify Myo5b as a motor protein that transports FN1-containing vesicles in myofibroblasts. Supporting this conclusion, we obtained the following findings: (1) FN1-containing vesicles and Myo5b show notable colocalization on actin filaments in myofibroblasts produced from HPMCs by MesoMT. (2) Proximity ligation assay that detects close association of Myo5b and FN1 within 50 nm of distance revealed that Myo5b and FN1 either directly or indirectly bind to each other in cells. (3) Myo5b KD by specific siRNA significantly decreases FN1 secretion. (4) Live cell imaging revealed Myo5b and FN1(EDA)-containing vesicles show continuous directional co-movement in HPMCs.

Myosin constitutes a large superfamily of more than 35 classes and 10 classes are found in vertebrates [17]. Myosin family members can be classified into two groups based upon their nature as a motor protein. One is the motor suitable to produce large force, such as myosin II. This type of myosin spends the majority of the cross-bridge cycle time dissociated from actin and only a short time associated with actin to produce power stroke to move actin [18]. This nature is suitable for producing a large force since many myosin molecules can interact with single actin filaments without interfering with each other. However, this group of myosins is not suitable for vesicular transportation since it cannot move continuously on actin filaments without dissociation. The other group of myosin spends a majority of cross-bridge cycling time associated with actin. This type of myosin, often called “processive” myosin, is a suitable motor for specific cargo transportation such as intracellular vesicles [18]. Class V myosin is typical processive myosin, therefore it is consistent that Myo5b, a member of class V myosin processively transports FN1 vesicles.

It has been shown that the motor activity of Myo5a, a member of class 5 myosin, is regulated through the intramolecular interaction between the globular tail domain and the motor domain [50]. In the inhibited conformation, Myo5a forms a folded conformation, in which the tail domain associates with the motor domain. The binding of the globular tail domain to the motor domain interferes with the cross-bridge cycle movement, thus functioning as an intramolecular inhibitor [20,51]. This tail-dependent inhibited conformation is altered to an active extended conformation when Myo5a binds to a specific binding protein, melanophilin, to the globular tail that releases this inhibition, thus activating the motor activity of Myo5a [20]. A similar scenario of the regulation of Myo5b was reported that the binding of Rab11a to the globular tail can in part attenuate the inhibition [52]. In the present study, we found that Rab11b co-localizes with FN1 and Myo5b in HPMCs. Proximity ligation assay revealed that TGF-β significantly increased the proximity ligation signal intensity between Rab11b and Myo5b indicating a close association between Myo5b and Rab11b (Figure 8). Moreover, the active mutant of Rab11b, i.e., Rab11b S20V but not the inactive mutant of Rab11b (Rab11b S25N) showed colocalization with Myo5b (Figure 7). These results suggest that Myo5b associates with Rab11b and TGF-β stimulation promotes the binding of these proteins (Figure 7, Figure 8, Figure 9 and Figure 10). It was suggested using a yeast two-hybrid assay that Myo5b tail domain may interact with Rab11b S20V (active mutant), but not Rab11b S25N (dominant negative mutant) [53]. The present results are consistent with that obtained by a yeast two-hybrid assay. Supporting this view, we found that TGF-β stimulation induces the formation of the active form of Rab11b (Figure 11). Further, Rab11b gene silencing attenuated the TGF-β induced secretion of FN1 (Figure 6). These results suggest that Myo5b binds to Rab11b, which plays an important role in the transportation and secretion of FN1 in HPMCs.

Quite interestingly, when Myo5b is overexpressed all three proteins, i.e., FN1, Rab11b and Myo5b, showed high colocalization (Figure 6). The result suggests that Myo5b strengthens the association of Rab11b with FN1 vesicles. In other words, Myo5b is not simply a binding partner of Rab11b, but it may function as a regulator of Rab11b to facilitate association of FN1 vesicles. We also found that the active mutant of Rab11b showed punctate localization that well colocalize with Myo5b while the dominant negative mutant was diffuse (Figure 7). These results are consistent with the above finding and suggest that Myo5b facilitates association of Rab11b with FN1, presumably due to stabilization of Rab11b-GTP form. This view was supported by the finding that Rab11b-GTPγS but not Rab11b-GDPβS directly binds to the Tail domain of Myo5b (Figure 11).

We found that Rab11b gene silencing significantly diminished the secretion of FN1 from HPMCs. The results are consistent with an earlier report that Rab11b down-regulation decreases FN1 secretion from cultured primary arterial endothelial cells [54]. On the other hand, gene silencing of Rab11a, isoform of the Rab11 family, did not significantly attenuate the secretion of FN1 from HPMCs. Although Rab11a and Rab11b share high amino acid homology (~90%), several reports suggested the distinct function of the two isoforms. Rab11a and Rab11b were localized at the different vesicle compartments in MDCK and gastric parietal cells [55]. Rab11b, but not Rab11a regulated cystic fibrosis transmembrane conductance regulator (CFTR) recycling to the apical membrane although both Rab11a and Rab11b were localized in the same vesicles containing CFTR [56]. The present findings are consistent with these earlier reports and support the distinct function of Rab11a and Rab11b. Based upon these findings, we concluded that Rab11b, but not Rab11a plays an important role in the transportation of FN1-containing vesicles in HMPCs.

It was previously reported that Rab11a partially activated the actin-activated ATPase activity of Myo5b in vitro, presumably due to the disruption of intra-molecular inhibitor function of the globular tail domain [52]. In the present study, we found that Rab11b-GTP binds to the tail domain of Myo5b, and significantly activates the actin-activated ATPase activity of Myo5b that is closely correlated with the movement activity of Myo5b. Importantly, the Rab11b induced activation of Myo5b is dependent on the activation of Rab11b, i.e., formation of Rab11b-GTP form, and TGF-β facilitates the activation of Rab11b in HPMCs. Since the active form of Rab11b directly binds to the globular tail domain of Myo5b, it is likely that Rab11b binding to the tail domain interferes with the intra-molecular interaction between the motor domain and the globular tail domain. Above results suggest that Rab11b activates Myo5b based transportation of FN1-containing vesicles by two folds, i.e., facilitating the association of Myo5b motor with the FN1-containing vesicles and the activation of the motor activity thus activating the movement of the vesicles.

Present study identified that Myo5b is an important motor for FN1 transportation and secretion, thus actin-based transportation system plays an important role in FN1 transportation and secretion. Proposed mechanism of Myo5b driven transportation of FN1-containing vesicles based upon the present findings is as follows. TGF-β stimulation activates Rab11b to form a GTP bound form. The activated Rab11b-GTP recruits Myo5b to the FN1-containing vesicles and activates the motor activity of Myo5b. Activated Myo5b continuously move the FN1-containing vesicles along with the actin filaments towards cell periphery, thus facilitating the secretion of FN from the cells. Future studies will involve the characterization of pleural fibrosis progression in mice with a targeted deletion of Myo5b. Floxed myo5b mice [57] will cross with calb2 cre-mice to generate mice with mesothelial cell specific loss of Myo5b, as we previously reported [49]. These studies will determine the effectiveness of Myo5b targeting on disease progression.

Our results indicate that Myo5b transports FN1 and facilitates secretion from transitioned HPMCs. It has been postulated that the directional transportation of intracellular vesicles is driven by both microtubule and microfilament transportation systems [58]. For instance, both kinesin and Myo5a are involved in the transportation and secretion of melanosomes from melanocytes [59]. For FN1 transportation, it is likely that microtubule motors such as kinesin family members contribute to such a movement. To date, nothing is known about possible microtubule based motors involved in FN1 transportation, and further study is required to understand entire transportation systems for FN1 secretion.

## 4. Materials and Methods

### 4.1. Mouse Disease Models

All experiments involving animals were approved by the Institutional Animal Care and Use Committee at the University of Texas Health Science Center at Tyler. All experiments relating to animals were performed in accordance with relevant guidelines and regulations [IACUC protocol numbers: 648 (Date of approval, 24 June 2019) and 689 (Date of approval, 17 March 2021)]. Wild-type C57BL/6j mice were intrapleurally treated with saline, carbon black/bleomycin (CBB) for 14 days or *S. pneumoniae* for 7d as previously described [21,60].

### 4.2. HPMC Isolation and Culture

Permission to collect and use HPMCs was approved by the Institutional Human Subjects Review Board of the University of Texas Health Science Center at Tyler [Protocol code, 2020-039 (Date of approval, 14 September 2020)]. All experiments regarding human subjects were performed in accordance with relevant guidelines and regulations. Cells were isolated from pleural fluids collected from patients with congestive heart failure or post coronary bypass pleural effusions [61] and maintained on dishes with CellBIND surface (Corning, New York, NY, USA) using LHC-8 culture medium (Thermo Fisher Scientific, Waltham, MA, USA) or BEGM (without retinoic acid and epinephrine, Lonza, Basel, Switzerland) containing 3% fetal bovine serum (Thermo Fisher Scientific), 2% antibiotic-antimycotic (Thermo Fisher Scientific), and 1% L-glutamine (Thermo Fisher Scientific) in a humidified incubator at 37 °C and 5% CO_2_/95% air.

### 4.3. Cellular Treatment

Cells were incubated in serum-free medium (SFM) of RPMI 1640 (HyClone, Logan, UT, USA) supplemented with GlutaMAX (Thermo Fisher Scientific) for 8 h prior to treatment with recombinant human TGF-β (5 ng/mL, R&D Systems, Minneapolis, MN, USA). Cells were then allowed to incubate for 24 h (Quantitative PCR analysis) or 48 h (Western blotting and immunostaining analysis) at 37 °C and 5 % CO_2_/95 % air.

### 4.4. Antibodies

The following antibodies were used for immunocytochemistry and Western blot analysis: anti-FN1 antibody (MAB1918, R&D Systems, 1:400 for immunocytochemistry, 1:2000 for Western blot), anti-FN1(EDA) antibody (IST-9, Santa Cruz, Dallas, TX, USA 1:2000 for Western blot), anti-Myo5b antibody (NBP1 87746, Novus, St. Louis, MO, USA 1:200 for immunocytochemistry, 1:2000 for Western blot). anti-β-actin antibody (AC-15, Sigma Aldrich, St. Louis, MO, USA, 1:5000 for Western blot). anti-α-SMA antibody (1A4, R&D Systems, 1:5000 for Western blot). anti-α-SMA antibody (17H19L35, Thermo Fisher Scientific, 1:400 for immunocytochemistry), anti-Collagen I antibody (1310-01, SouthernBiotech, Birminghum, AL, USA, 1:400 for immunocytochemistry), anti-RAB11a antibody (71-5300, Thermo Fisher Scientific, 1:3000 for Western blot), anti-RAB11b antibody (19742-1-AP, Thermo Fisher Scientific, 1:2000 for Western blot). anti-Flag antibody (F1804, Sigma Aldrich, 1:3000 for Western blot). anti-V5-antibody (V8012, Sigma Aldrich, 1:1000 for immunocytochemistry, 1:3000 for Western blot).

### 4.5. Quantitative PCR

Total mRNA was isolated using RNeasy Mini kit (Qiagen, Hilden, Germany) according to the manufacturer’s instructions. Total mRNA was then reverse-transcribed into total cDNA using the SuperScript VILO (Thermo Fisher Scientific) according to the manufacturer’s instructions. Quantitative PCR analysis was then performed on the cDNA using QuantStudio 6 Flex (Applied Biosystems, Waltham, MA, USA) and Taqman Assays for Myo5a (Hs00165309_m1), Myo5b (Hs00393037_m1), Rab11a (Hs00366449_g1), Rab11b (Hs00188448_m1), α-SMA (Hs00426835_g1) and FN1 (Hs01549976_m1) (Applied Biosystems). GAPDH (Applied Biosystems) was used as a loading control.

### 4.6. Plasmid DNA

Human Myo5b-, Rab11a-, Rab11b-, Rab11FIP3 cDNA were purchased from DNASU (Arizona State University, AZ; HsCD00744481, HsCD00632673, HsCD00632681, HsCD00946210). FN1-EDA cDNA was purchased from GenScript, Piscataway, NJ, USA (NM 212482.3).

### 4.7. Histochemistry and Immunofluorescence Staining of Mice Tissues

Tissues sections were first deparaffinized and subjected to antigen retrieval using a citrate buffer at 95 °C for 20 min as previously described [29]. Immunostaining was performed by using antibodies of Col-1 (1310-01, SouthernBiotech), α-SMA (17H19L35, Thermo Fisher Scientific) and FN1 antibody (MAB1918, R&D Systems). Briefly, mouse tissue sections were blocked using a proprietary blocking solution from a M.O.M. kit (Vector Laboratories, Burlingame, CA, USA). Primary antibodies were then incubated overnight at 4 °C in kit diluent. First antibodies were visualized with Alexa Fluor 488, 568 and 647 secondary antibodies (Life Technologies, Carlsbad, CA, USA), and nuclei were stained with Hoechst 33342 (Thermo Fisher Scientific). Tissues were mounted onto slides with Fluoro Gel with DABCO (Electron Microscopy Sciences, Hatfield, PA, USA). Tissue staining images were taken by Leica TCS SP8 systems (Leica Microsystems, Wetzlar, Germany).

### 4.8. Western Blotting

Conditioned media (CM) were collected, and cells were washed with PBS and lysed with 200 µL PBS containing 1% NP40, 20 mM Tris-HCl (PH 7.5), 137 mM NaCl, 1 mM Na_2_VO_4_, 1 mM EDTA supplemented with Halt Protease Inhibitor Cocktail (Thermo Fisher Scientific). Cell lysates were then incubated on ice for 30 min, and debris was removed by centrifuging. Protein concentrations of cell lysates were determined using Pierce BCA assay kit (Thermo Fisher Scientific). Immunoblots were then imaged by Molecular Imager ChemiDoc XRS+ (Bio-Rad, Hercules, CA, USA).

### 4.9. Immunofluorescence Staining

Cells were washed with PBS, placed in fixation solution (4 % formaldehyde, 2 mM MgCl_2_, and 1 mM EGTA in PBS), washed with PBS, and permeabilized with 0.05% Triton X-100 in PBS for 10 min. Cells were then washed and blocked with 5% BSA for 1 h. First antibodies were diluted with 5% BSA, then applied to cells and incubated overnight and visualized with Alexa Fluor 488, 568 and 647 secondary antibodies (Thermo Fisher Scientific), and nuclei were stained with Hoechst 33342 (Thermo Fisher Scientific). Proximity ligation assay was performed according to the Olink Bioscience protocol using Duolink In situ Red starter kit (Sigma Aldrich). After incubation with primary antibodies as described above, cells were washed and incubated with PLA probes (anti-mouse MINUS and anti-rabbit PLUS) for 1 h at 37°C. Subsequent ligation and detection were performed according to the manufacturer’s protocol. Cells were mounted onto slides with Fluoro Gel with DABCO (Electron Microscopy SciencesFluorescence images were taken by Leica TCS SP8 systems (Leica Microsystems) for confocal microscopy.

### 4.10. Gene Silencing with Specific siRNA

Cells were transfected with 20 nM of Myo5A predesigned siRNA (SASI_Hs01_00181251; Sigma Aldrich), Myo5B predesigned siRNA (SASI_Hs01_00220872; Sigma Aldrich), RAB11A predesigned siRNA (SASI_Hs01_00126206; Sigma Aldrich), RAB11B predesigned siRNA (SASI_Hs01_00220872; Sigma Aldrich) or Negative control siRNA (SIC002; Sigma Aldrich) for 5 h at 37 °C in SFM using Lipofectamine RNAiMAX (Thermo Fisher Scientific), then were maintained in LHC8 complete medium and used in the following experiments.

### 4.11. Live Cell Imaging

GFP-Myo5a and mCherry-FN1 was cloned in BacMam pCMV-DEST Vector using the Gateway cloning system. For GFP-Myo5a, GFP tag was cloned to *N*-terminal of Myo5a. For mCherry-FN1, mCherry-tag was cloned between FN1-III domains 3 and 4. The sequence at the insertion site was (FN1-III 3).TTGTGGRMVSK. (mCherry). ELYKGGRPRSD. (FN1-III 4) [the mCherry sequence is underlined; the *Not* I restriction site added three extra amino acids (GGR) at each end of the mCherry] [62]. The recombinant baculoviruses expressing GFP-Myo5a and mCherry-FN1 were produced using ViraPower BacMam Expression System (Thermo Fisher Scientific) according to manufacture’s protocol. Live cell imaging was performed by using DeltaVision OMX (GE Healthcare Life Sciences, Chicago, IL, USA) in 48 h after transfection.

### 4.12. Protein Purification

To express the GST-Myo5b Tail domain (1440−1848 a.a.) and GST-Rab11FIP3 Rab binding domain (GST-FIP3 RBD; 694−756 a.a), both cDNAs were amplified by PCR with each single set of primers containing restriction sites and subcloned into the pET-30a vector (Novagen, Pretoria, South Africa) having GST tag at *N*-terminal and 6xHis tag at *C*-terminal. These plasmids were transfected into *Escherichia coli* BL21 (DE3) and the cells were grown in 2YT media to an OD600 of 0.5, and then protein expression was induced with 0.1 mM IPTG for 16 h at 23 °C. Pellets were lysed with sonication in Lysis Buffer [200 mM NaCl, 50 mM HEPES-KOH (PH7.5), 0.5% NP40, 5 mM β-mercaptoethanol, 5 mM Imidazole, 2 mM MgCl_2_, 0.2 mM EGTA, 1 mM phenylmethylsulfonyl fluoride, 2 µg/mL pepstatin A, and 1 µg/mL trypsin inhibitor]. After centrifugation at 100,000× *g* for 20 min, the supernatant was mixed with nickel-nitrilotriacetic acid (Ni-NTA) agarose (Qiagen) and gently rotated for 1 h at 4 °C. After extensive wash with Lysis buffer plus 25 mM imidazole, the resins were packed to a Poly-prep chromatography column (Bio-Rad), and the protein was eluted with elution buffer containing 200 mM NaCl, 10 mM HEPES-KOH (PH7.5), 1 mM DTT, 200 mM imidazole and 10 % sucrose.

Full length of Myo5b heavy chain and full length Rab11b cDNAs were transferred to pFastbac HT (Thermo Fisher Scientific) that is modified to express FLAG Tag at the *N*-terminal side of cDNA of interest. Approximately 2 × 10^9^ Sf9 cells (Thermo Fisher Scientific,) were infected with Myo5b heavy chain plus calmodulin light chain or Rab11b expressing baculoviruses. The cells were harvested at 72 h after infection.

For Myo5b, the cells were pelleted by centrifugation, and homogenized in a lysis buffer (0.1 M KCl, 50 mM Tris-HCl (pH7.5), 2 mM MgCl_2_, 5 mM EGTA, 1 mM ATP, 1 mM DTT, and 10 µg/mL leupeptin). The supernatant by ultracentrifugation was mixed with anti-FLAG antibody-agarose resins (Sigma Aldrich) and the tube containing the suspension was rotated at 4 °C for 1 h. After washed with a buffer (0.2 M NaCl, 20 mM MOPS-KOH (pH 7.5), 1 mM EGTA, 1 mM DTT, 10 µg/mL leupeptin), the resins were packed to a Poly-prep chromatography column, and the protein was eluted with the elution buffer (the wash buffer containing 0.1 mg/mL FLAG peptide and 10 % sucrose). Purified protein was concentrated with Vivaspin 500 (30 kDa MWCO, Sartorius, Göttingen, Germany).

For Rab11b, the Sf9 cell pellet was homogenized in PBS, 1 mM DTT, and 10 µg/mL leupeptin. The supernatant was cleared by ultracentrifugation, mixed with Ni-NTA resins and the tube containing the suspension was rotated at 4 °C for 1 h. After two washes with 0.3 M NaCl and 10 mM Imidazole-HCl (pH7.5), the resins were packed to a column, and the protein was eluted with 0.3 M NaCl, 200 mM Imidazole-HCl (pH7.5) and 10% sucrose. Purified protein was concentrated with a Vivaspin 6 (10 kDa MWCO, Sartorius). All purified proteins were snap-frozen in liquid nitrogen and stored at −80 °C.

### 4.13. GST-Pull down Assay

The purified flag-Rab11b was pretreated in buffer (20 mM Tris-HCl PH 7.5, 150 mM NaCl, 1 mM MgCl_2_, 1 mM EGTA, 1 mM DTT) in the presence of 5-fold excess of GTPγS, or GDPβS for 30 min at 37 °C. In other assays flag-Rab11b was added to the lysates of cells treated with or without TGF-β. Thirty microliters of 10 µM Rab11b and 2.5 µM Myo5b Tail in binding buffer (20 mM Tris-HCl PH 7.5, 150 mM NaCl, 1 mM MgCl_2_, 1 mM EGTA, 1 mM DTT, 0.05 % Tween, 1% BSA) in the presence of 20 µM GTPγS, GDPβS, or absence of the any analogs were mixed with 5 µL of Glutathione Sepharose 4B (GE Healthcare Life Sciences) and incubated with rotation at 4 °C for 1 h. The beads were collected by brief centrifugation and washed 3 times with 300 ul of buffer (20 mM Tris-HCl PH 7.5, 150 mM NaCl, 1 mM MgCl_2_, 1 mM EGTA, 1 mM DTT, 0.05 % Tween). The bound proteins were eluted with 25 µL of 20 mM Glutathione in Tris-HCl (PH8.0), 100 mM NaCl and 1 mM DTT. The eluted proteins were subjected to SDS-PAGE. GST-Myo5b Tail was visualized by Coomassie Brilliant Blue staining. Flag-Rab11b were analyzed by Western blot using flag antibody.

### 4.14. ATPase Assay

Rab11b was pre-incubated at 37 °C for 30 min in the presence of GDPβS or GTPγS. Rab11b (15 µM) was then mixed with 5.3 µg/mL (20 nM) Myo5b in a buffer containing 0.1 M KCl, 20 mM MOPS-KOH (pH 7.5), 3 mM MgCl_2_, 2 mM ATP, 10 µM CaM, 20 unit/mL pyruvate kinase, 2.5 mM phosphoenolpyruvate, 1 mM EGTA, and 20 µM F-actin. The assay of ATPase activity was performed at 37 °C. Liberated pyruvate was determined as described previously [63].

## 5. Conclusions

Present findings indicate that Myo5b and Rab11b play a key role in the transportation and secretion of FN1 from HPMCs. We conclude that TGF-β stimulation activates Rab11b to form Rab11b-GTP, which facilitates the association of Rab11b to FN1-containing vesicles as well as the binding of Ran11b to the tail domain of Myo5b. The bound Rab11b-GTP activates the motor activity of Myo5b, and the activated Myo5b continuously moves FN1-containing vesicles to cell peripheries and facilitates FN1 secretion from the cells. The present findings support that Myo5b and Rab11b may contribute to the development of pleural fibrosis.

## Figures and Tables

**Figure 1 ijms-23-04823-f001:**
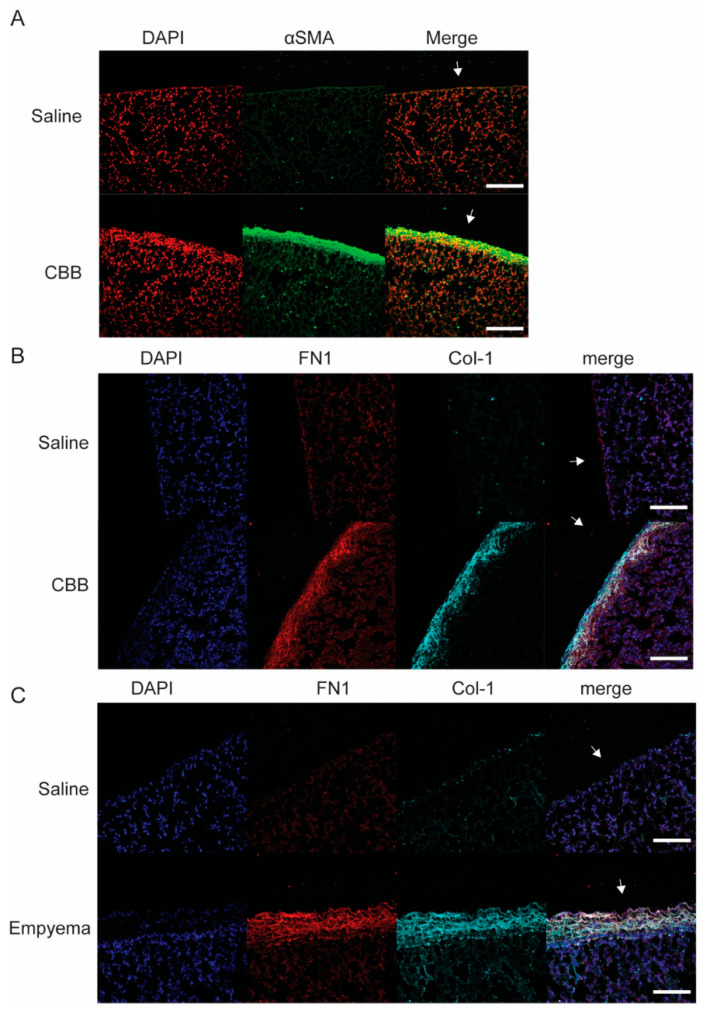
Increased fibronectin (FN1) deposition in two fibrosing pleural injury models in mice: Carbon black bleomycin (CBB) and empyema. (**A**) Lung tissues sections from saline and CBB-treated mice were stained for α-smooth muscle actin (α-SMA; green) and nuclei (red). α-SMA were localized at the pleura in CBB-treated mice. Representative image of *n* = 3 tissues. (**B**) Lung tissues sections from saline and CBB-treated mice were stained for FN1 (red), Collagen 1 (Col-1; cyan) and nuclei (blue). FN1 and Col-1 were colocalized at the pleura in CBB-treated mice. Representative image of *n* = 5 tissues. (**C**) Lung tissue sections from saline and empyema -treated mice were stained for FN1 (red), Col-1 (cyan) and nuclei (blue). FN1 and Col-1 were colocalized at the pleura in empyema lung tissues. Representative image of *n* = 5 tissues. White arrows indicate pleura. Scale bars; 100 μm.

**Figure 2 ijms-23-04823-f002:**
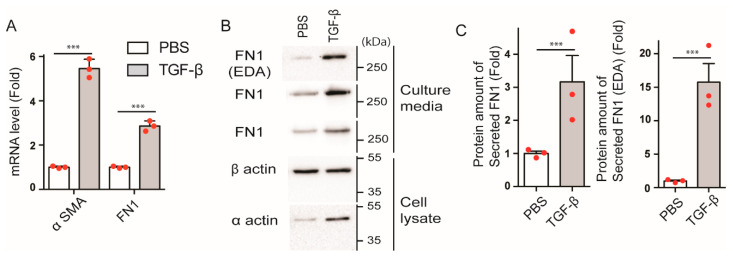
Effect of TGF-β on FN1 expression and secretion from HPMCs. (**A**) mRNA levels of FN1 in HPMCs were determined by q-PCR analysis. The cells were serum-starved for 8 h and treated with PBS or TGF-β (5 ng/mL) for 48 h. Expression levels were normalized to the control (PBS). *n* = 3. *** *p* < 0.001 vs. PBS. (**B**) Representative Western blot. Full-length blots were presented in Appendix A. (**C**) Statistical representation of Western blot analysis. *n* = 3, *** *p* < 0.001 vs. PBS. FN1 protein expression levels were analyzed by Western blotting in both cell lysate and culture medium. Culture media were changed at 48 h after TGF-β stimulation and the culture media for Western blotting were collected after 24 h. Cell lysate were prepared after the collection of the culture medium. The red spots in bar graphs display the distribution of data points.

**Figure 3 ijms-23-04823-f003:**
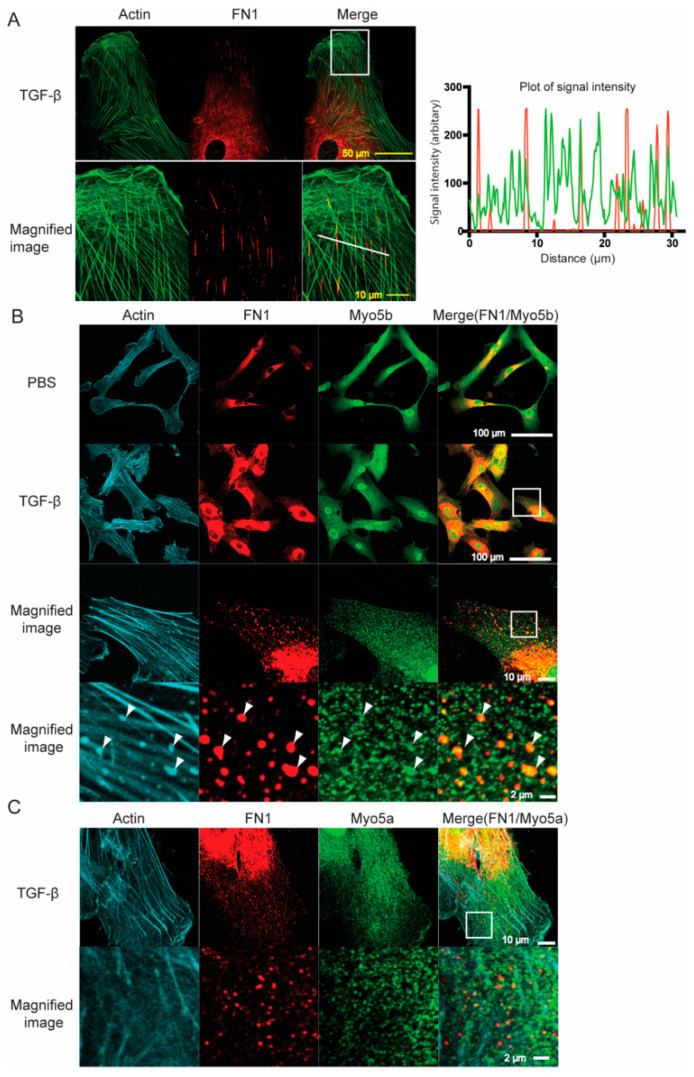
Colocalization of Myo5 isoforms and FN1 in HPMCs. (**A**) Confocal images of FN1 and actin in HPMCs after TGF-β stimulation. Left panel, confocal images. Bottom panels are magnified images of the white box of the top panel; right panel, signal intensities of FN1 (red) and actin (green) of the white line in the left panel. Distance 0 represents the left end of the white line. Note that FN1 showed apparent filamentous localization that coincided with actin filaments. Representative image of *n* = 6 cells. (**B**) Confocal images showing the localization of endogenous FN1 (red) and endogenous Myo5b (green) in HPMCs in the absence and presence of TGF-β (5 ng/mL) for 48 h. TGF-β stimulation increased the signal intensity of FN1. DAPI (blue), actin (cyan). Scale bars, 100 µm. Middle panels: Magnified images correspond to the white square of upper panel. Bottom panels: Further magnified image that corresponds to the white square area of the middle panel. Myo5b was colocalized with FN1-containing vesicles (indicated by arrow heads) along with actin fibers. Scale bar, upper panels: 100 μm, middle pannels: 10 μm, lower panel: 2μm. Representative images of *n* = 3 cells for PBS, *n* = 6 cells for TGF-β. (**C**) Confocal images showing the localization of endogenous FN1 (red) and endogenous Myo5a (green) in HPMCs in the presence of TGF-β (5 ng/mL) for 48 h. Myo5a was not notably colocalized with FN1 after TGF-β stimulation. Scale bar, Upper: 10 μm, Lower: 2 μm. Representative image of *n* = 3 cells.

**Figure 4 ijms-23-04823-f004:**
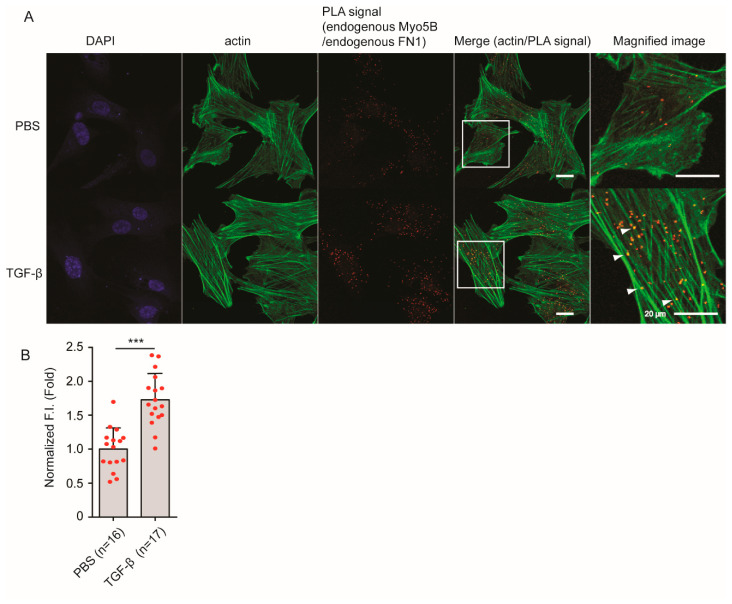
TGF-β stimulation increased close association between Myo5b and FN1 revealed by Proximity-ligation assay. (**A**) TGF-β stimulation increased proximity-ligation (PLA) signal between Myo5b and FN1. Scale bars, 20 μm. Magnified images of the boxed areas are shown in the right panels. Proximity ligation signals are found on actin filaments (arrowheads). Scale bars, 20 μm. (**B**) Quantitative analysis of intensity of the signals (mean ± SD), *** *p* < 0.001. Proximity-ligation assay revealed close association of Myo5b and FN1 that is induced by TGF-β stimulation. The red spots in bar graph displays the distribution of data points.

**Figure 5 ijms-23-04823-f005:**
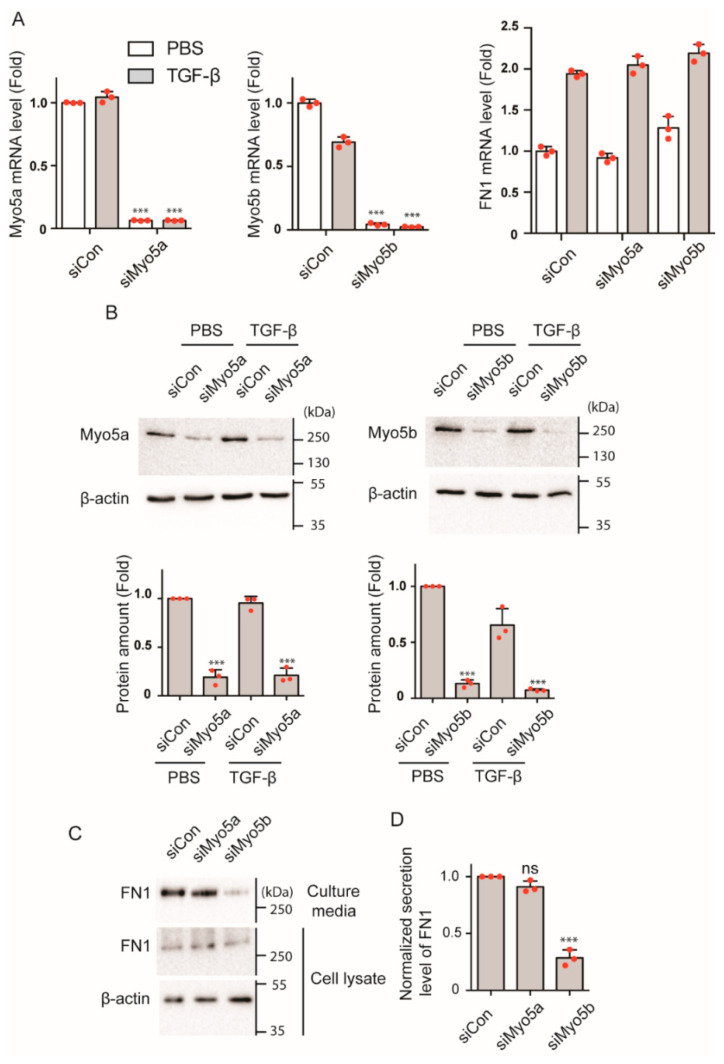
Effect of Myo5b KD on secretion of FN1 from HPMCs. (**A**) Effect of gene silencing of Myo5 isoforms on mRNA level of Myo5 and FN1 by specific siRNAs. mRNA expression was examined by qPCR analysis. siCon, control siRNA; siMyo5a, Myo5a specific siRNA; siMyo5b, Myo5b specific siRNA. *n* = 3, *** *p* < 0.001 vs. siCon. (**B**) Effect of gene silencing of Myo5 isoforms on Myo5 protein expression level in cell lysate. Top, representative Western blots; Bottom, statistical representation of Western blot signal intensities (*n* = 3, mean ± SD). *** *p* < 0.001 vs. siCon. Full-length blots were presented in Appendix A. (**C**) Effect of Myo5 KD on FN1 secretion. Representative image of Western blot analysis of FN1 protein amount in cell lysate and culture media. Control siRNA and Myo5a or Myo5b specific siRNA transfected HPMCs were serum starved for 8 h and cells were then treated with TGF-β (5 ng/mL). The cell lysate and culture media were collected at 4 days after TGF-β stimulation. The culture media were changed at 24 h before collecting the culture media for determination of secreted FN1 and the culture supernatants were subjected to Western blot analysis. Full-length blots were presented in Appendix A. (**D**) Statistical representation of normalized secreted level of FN1 after TGF-β stimulation (*n* = 3, mean ± SD). *** *p* < 0.001. ns, not significant. The red spots in bar graphs display the distribution of data points.

**Figure 6 ijms-23-04823-f006:**
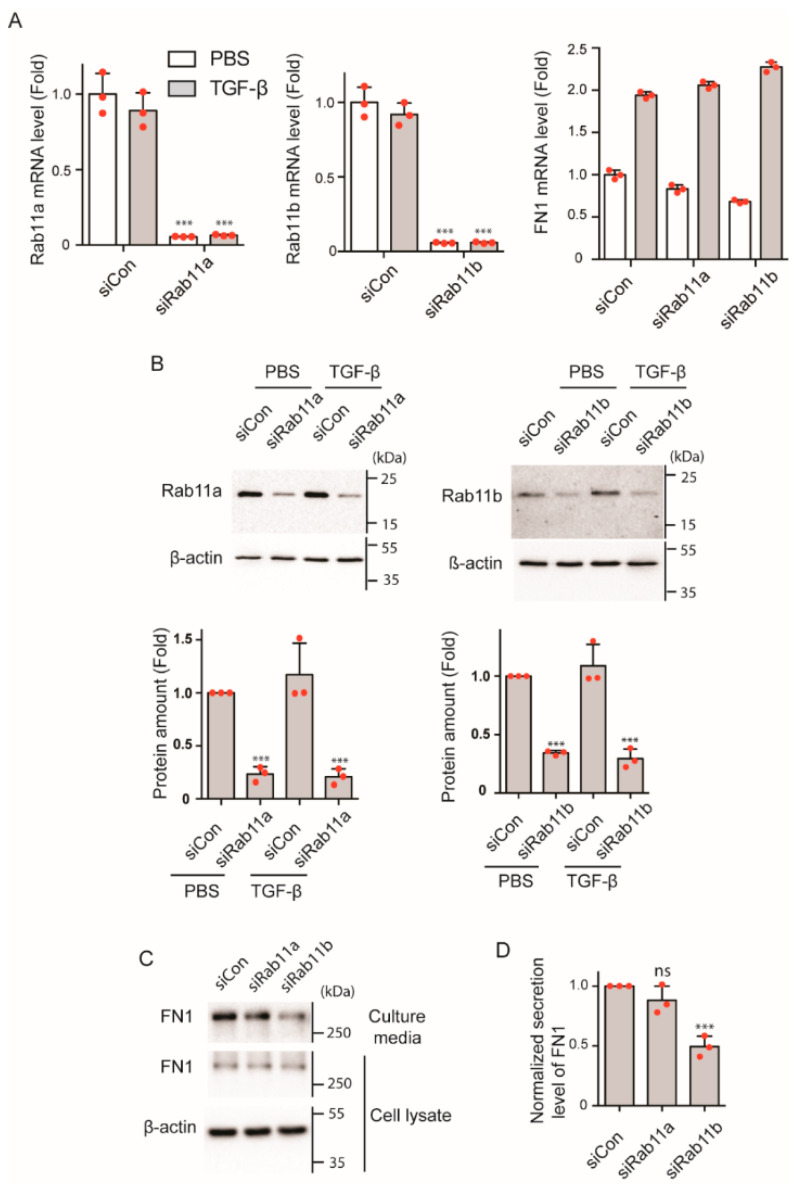
Effect of gene silencing of Rab11a and Rab11b, respectively on FN1 secretion. (**A**) Effect of gene silencing of Rab11 isoforms on mRNA level of Rab11 isoforms and FN1 by specific siRNAs. mRNA expression was examined by qPCR analysis. siCON, control siRNA; siRab11a, Rab11a specific siRNA; siRab11b, Rab11b specific siRNA. *** *p* < 0.001 vs. siCon. ns, not significant (*n* = 3, mean ± SD). (**B**) Effect of Rab11 gene silencing on Rab11 protein expression level in cell lysate. Top, representative Western blot; Bottom, statistical representation of the Western blot signal (*n* = 3, mean ± SD). *** *p* < 0.001 vs. siCon. Full-length blots were presented in Appendix A. (**C**) Effect of Rab11 isoform KD on FN1 secretion. Representative image of Western blot analysis of FN1 protein amount in cell lysate and culture media. Control siRNA and Rab11a or Rab11b specific siRNA transfected HPMCs were serum starved for 16 h and cells were then treated as described in the legend for Figure 5. Full-length blots were presented in Appendix A. (**D**) Statistical representation of normalized secreted level of FN1 after TGF-β stimulation (*n* = 3, mean ± SD). *** *p* < 0.001. ns, not significant. The red spots in bar graphs display the distribution of data points.

**Figure 7 ijms-23-04823-f007:**
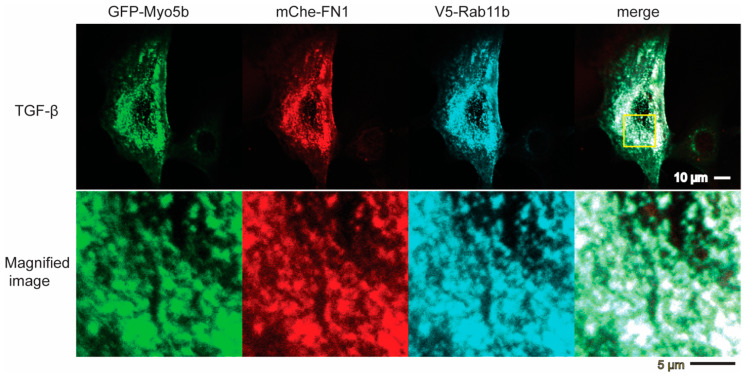
Colocalization of Myo5b, Rab11b and FN1. Confocal images of localization of GFP-Myo5b, mCherry-FN1 and V5-Rab11b in HPMCs after TGF-β stimulation. GFP- Myo5b (green), mCherry-FN1(red) and V5-Rab11b (cyan) were expressed in HPMCs and subjected to confocal microscopy. After fixation, the cells were stained with anti-GFP, anti-v5 antibodies, respectively. Bottom panels are enlarged images of the boxed area in the top. Scale bar, Upper: 10 μm, Lower: 5μm. Representative image of *n* = 6 cells.

**Figure 8 ijms-23-04823-f008:**
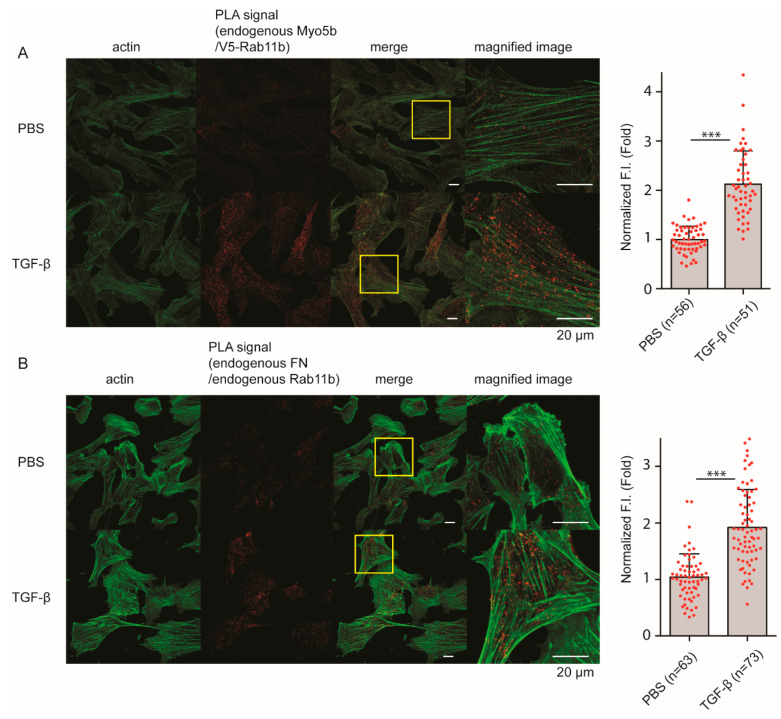
TGF-β increases Proximity ligation signal between Myo5b and Rab11b, and FN1 and Rab11b, respectively. Confocal images of HPMCs with proximity ligation assay. Proximity ligation signals were shown in red. (**A**) Effect of TGF-β stimulation on Proximity ligation signal between Myo5b and V5-Rab11b. V5-Rab11b was expressed in HPMCs for 16 h and proximity ligation assay was performed. Endogenous Myo5b and V5-Rab11b were stained with anti-Myo5b- and anti-v5-antibodies, respectively. Left, Representative confocal images. Magnified images of the boxed area are shown in the right panels. Right, Statistical representation (mean ± SD). *** *p* < 0.001. (**B**) Effect of TGF-β stimulation on Proximity ligation signal between endogenous FN1 and endogenous Rab11b. Endogenous Rab11b and FN1 were stained with specific antibodies, respectively. Left, Representative confocal image. Magnified images of the boxed area are shown in the right panels. Right, Statistical representation. Scale bar; 20 μm. The red spots in bar graphs display the distribution of data points.

**Figure 9 ijms-23-04823-f009:**
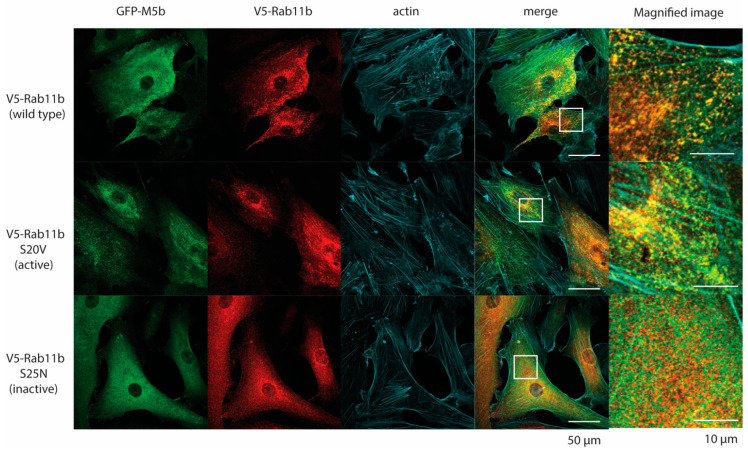
Effect of Rab11b activation on colocalization of Rab11b with Myo5b. Confocal images of localization of GFP- Myo5b and V5-Rab11b variants in HPMCs. GFP- Myo5b was expressed with v5-Rab11b (wild type), v5-Rab11b-S20V (active mutant) and v5-Rab11b-S25N (inactive mutant), respectively. After fixation, the cells were stained with anti-GFP- and anti-v5-antibodies, respectively. Magnified images of the boxed area are shown in the right. Note that the inactive variant showed diffuse localization. Representative images of *n* = 6 cells for each Rab11b constructs.

**Figure 10 ijms-23-04823-f010:**
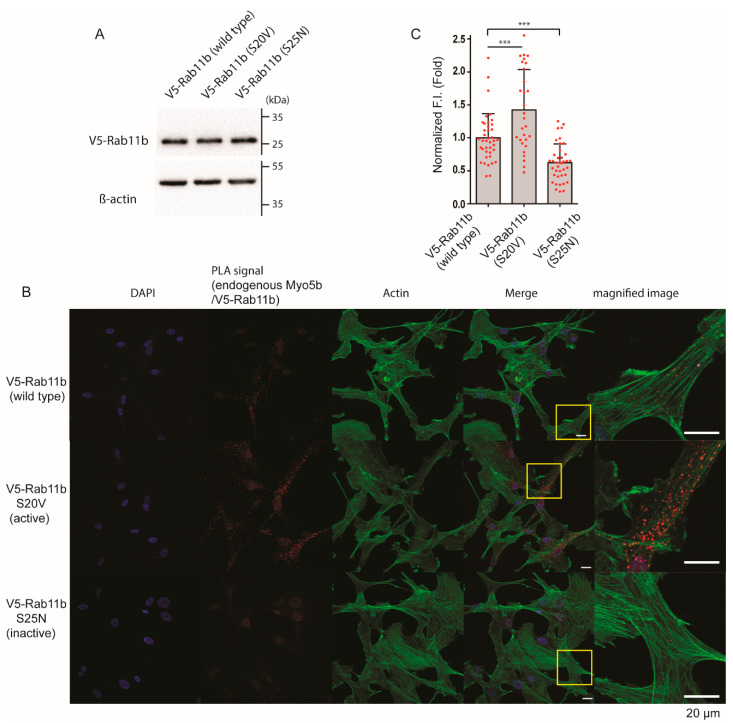
Effect of Rab11b activation on close association between Myo5b and Rab11b revealed by Proximity ligation assay. (**A**) Western blot of HPMCs expressing v5-Rab11b, v5-Rab11b-S20V and v5-Rab11b-S25N, respectively. V5-Rab11b variants were expressed in HPMCs at comparable levels. Full-length blots were presented in Appendix A. (**B**) Confocal images of proximity ligation assay of Myo5b and Rab11b variants in HPMCs. V5-Rab11b variants were expressed in HPMCs and proximity ligation assay was performed. Proximity ligation signals were shown in red. Scale bars; 20 μm. (**C**) Statistical representation of the proximity ligation signal intensities between endogenous Myo5b and each V5-Rab11 variants (mean ± SD). *** *p* < 0.001. The red spots in bar graphs display the distribution of data points.

**Figure 11 ijms-23-04823-f011:**
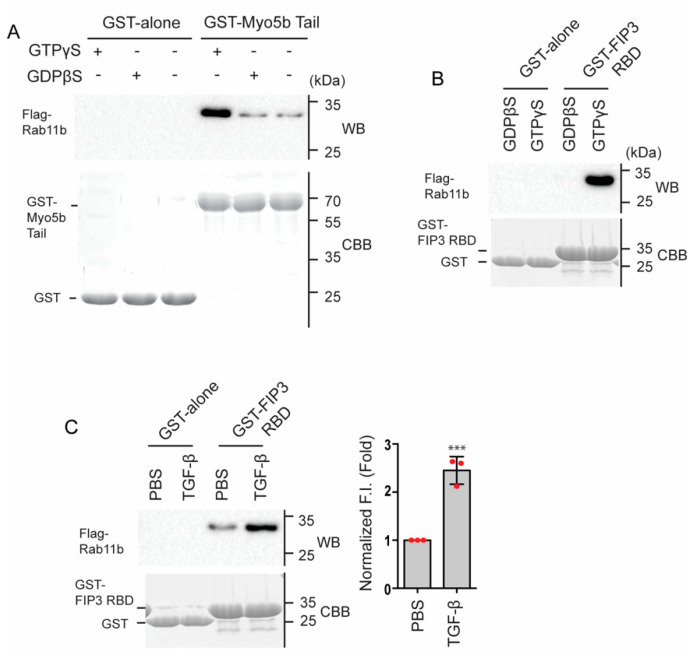
Effect of TGF-β on the activation of Rab11b in HPMCs. (**A**) GST-pull down assay showing the binding of GST- Myo5b Tail (M5b) (853−891 a.a.) to Flag-Rab11b. The pulled down sample with GST-beads were subjected to SDS page and visualized with CBB or Western blotting (WB) using anti-flag antibody. Pre-incubated Rab11b with GTPγS, but not with GDPβS enhanced the binding to GST-M5b Tail. Full-length blots were presented in Appendix A. (**B**) GTP binding to Rab11b enhances the binding with GST-FIP3-RBD (Rab11 binding domain). The pulled down proteins with GST-beads were subjected to SDS page and visualized by CBB or Western blotting using anti-flag antibody. Full-length blots were presented in Appendix A. (**C**) TGF-β stimulation activated Rab11b in HPMCs. HPMCs were stimulated with or without TGF-β. The cell lysates were incubated with GST-FIP3-RBD or GST alone, then subjected to GST pull-down assay. The pulled down proteins were subjected to SDS page and visualized with CBB or Western blotting using anti-Rab11b antibody. Left, representative western blot; Right, statistical representation of Western blot signal intensities of Flag-Rab11b (*n* = 3, mean ± SD). *** *p* < 0.001 vs. PBS. Full-length blots were presented in Appendix A. The red spots in bar graph displays the distribution of data points.

**Figure 12 ijms-23-04823-f012:**
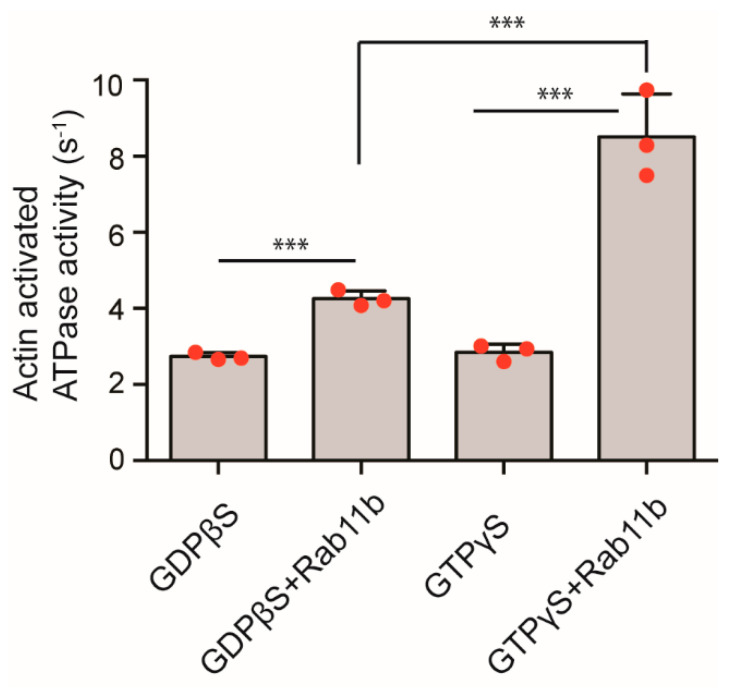
Effect of Rab11b on actin activated ATPase activity of Myo5b. Rab11b and GDPβS or GTPγS were pre-incubated at 37 °C for 30 min, and the ATPase activity of Myo5b with 20 µM F-actin in 1 mM EGTA at 37 °C was measured in the presence or absence of 15 µM Rab11b as described in “Materials and Methods” (*n* = 3, mean ± SD). *** *p* < 0.001. The red spots in bar graph displays the distribution of data points.

**Figure 13 ijms-23-04823-f013:**
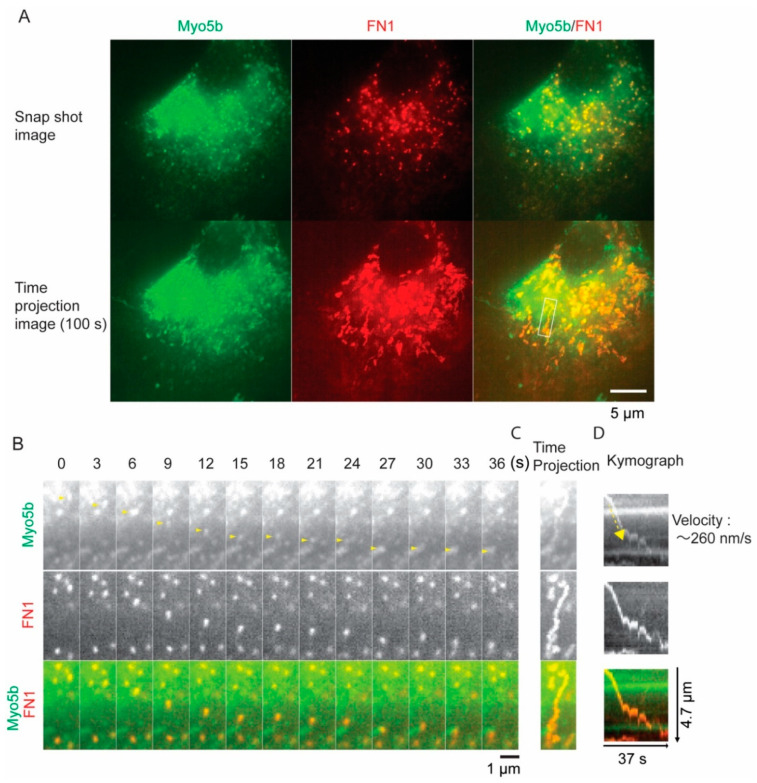
Direct visualization of the movement of Myo5b and FN1 in live HPMCs. (**A**) Representative first snap-shot image (top panel) and time projection images (bottom panel) (100 s; 100 frames) of GFP-Myo5b and mCherry-FN1(EDA) in live HPMC. Scale bar, 5 μm. Continuous movements are observed in the bottom panels. (**B**) Magnified time-lapse montages of the fluorescent proteins in white box in A. top, GFP-Myo5b; middle, mCherry-FN1 (EDA); bottom, merged images. Time (sec) is shown in the top. Scale bars, 1 μm. (**C**) Time projection image of (**B**). (**D**) kymograph showing the co-movement of GFP- Myo5b and mCherry-FN1 (EDA). The kymograph was generated based on the tracking line connecting the center of the fluorescent spot of GFP- Myo5b shown in yellow arrowheads in B. The velocity of the movement of GFP- Myo5b (yellow dotted line and arrow) is about 260 nm/s. Scale bar, 1 μm. HPMCs were stimulated with TGF-β, and then GFP- Myo5b and mCherry-FN1(EDA) were expressed using the corresponding viral expression vectors. The cells were imaged using DeltaVision OMX (GE Healthcare Life Sciences) at 10 h and 16 h after transduction for GFP-Myo5b and mCherry-FN1(EDA), respectively. Representative movie of *n* = 9 cells.

## Data Availability

The data presented in this study are available within the article text and figures.

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
