# Peer review of "Myo5b Transports Fibronectin-Containing Vesicles and Facilitates FN1 Secretion from Human Pleural Mesothelial Cells"

_ijms, 2022, doi:10.3390/ijms23094823_

Round 1
Reviewer 1 Report
The manuscript entitled “Myo5b transports fibronectin containing vesicles and facilitates FN1 secretion from human pleural mesothelial cells” explains the mechanism-of FN1 transportation and secretion. The manuscript is well written and the data looks clear to me.
I have only minor comments
- Expand the discussion and conclusion section by adding more points on future perspectives of this study.
- Discuss the importance of the study from the clinical perspective.
- Include a graphical abstract/scheme to summarise your study.
Reviewer 2 Report
In the current manuscript, the authors describe their findings regarding the role of Myo5b and Rab11b in fibronectin transportation in pleural mesothelial cells. They used an array of techniques, which were well chosen, yielded convincing results and are well presented.
The overall conclusions appear warranted by their results and constitute a significant addition to our knowledge in the field.
Author Response
Response to Reviewer 2 Comments
Reviewer 2 Comments: In the current manuscript, the authors describe their findings regarding the role of Myo5b and Rab11b in fibronectin transportation in pleural mesothelial cells. They used an array of techniques, which were well chosen, yielded convincing results and are well presented.
The overall conclusions appear warranted by their results and constitute a significant addition to our knowledge in the field.
We thank the reviewer for reading our paper and his encouraging statement.